# Determination of key functional structures of an amorphous VHL-based SMARCA2 PROTAC

Daria Torodii [1,8], Jacob B. Holmes [1,2,8], Manuel Cordova[1,2], Pinelopi Moutzouri[1], Lotte van Beek [3], Fredrik Edfeldt[4], Erik Malmerberg [5], Stig D. Friis [5], Johan R. Johansson [5], Alexander G. Milbradt[3], Sten O. Nilsson Lill[6], Benjamin Malfait[7], Staffan Schantz [7] ✉ & Lyndon Emsley [1,2] ✉

Proteolysis targeting chimeras (PROTACs) enable degradation of disease-related proteins via E3 ligase recruitment. PROTACs often do not easily crystallize, and they are usually formulated in amorphous forms. Determining the key interactions that stabilize the solid drug forms is of high interest. Here, we determine the complete atomic-level structure of an amorphous Von Hippel-Lindau (VHL)-based SMARCA2 PROTAC (PROTAC 2) using nuclear magnetic resonance (NMR) crystallography. We find that PROTAC 2 is more disordered as compared to previously studied amorphous formulations, and that the three functional units of the molecule have distinct structural types. In contrast to smaller drug molecules, where intermolecular hydrogen bonding interactions were found to be the main stabilization mechanism for the amorphous solid form, for PROTAC 2 we postulate that, in analogy to glassy polymers, the main stabilization mechanism is the entropic contribution introduced by the overall flexibility, especially in the linker region of the molecule. We also note that the most populated conformations found in the amorphous form differ from those of bound PROTAC 2 in the ternary protein complex as determined via X-ray crystallography. Our results provide insight into key structural features that stabilize amorphous formulations, specifically for molecules that can target proteins previously considered undruggable.

Proteolysis targeting chimeras (PROTACs) are heterobifunctional degraders used for targeted protein degradation, an emerging therapeutic modality with more than 20 compounds taken into clinical trials, and which is of intense current interest since they can in principle target proteins that were previously considered undruggable[1–11].

PROTACs contain two ligands that are covalently bound through a linker. One ligand has high affinity to a protein involved in, for example, the formation or proliferation of tumors or another disease-relevant protein, while the other ligand recruits and binds an E3 ubiquitin ligase. This structural bifunctionality ensures the proximity

[1]Institut des Sciences et Ingénierie Chimiques, École Polytechnique Fédérale de Lausanne (EPFL), Lausanne, Switzerland. [2]National Centre for Computational Design and Discovery of Novel Materials MARVEL, École Polytechnique Fédérale de Lausanne (EPFL), Lausanne, Switzerland. [3]Protein Science, Structure & Biophysics, Discovery Sciences, AstraZeneca R&D, Cambridge, UK. [4]Mechanistic and Structural Biology, Discovery Sciences, R&D, AstraZeneca, Gothenburg, Sweden. [5]Medicinal Chemistry, Research and Early Development Cardiovascular, Renal and Metabolism, BioPharmaceuticals R&D, AstraZeneca, Gothenburg, Sweden. [6]Data Science & Modelling, Pharmaceutical Sciences, R&D, AstraZeneca, Gothenburg, Sweden. [7]Oral Product Development, Pharmaceutical Technology & Development, Operations, AstraZeneca, Gothenburg, Sweden. [8]These authors contributed equally: Daria Torodii, Jacob B. Holmes. ✉e-mail: Staffan.Schantz@astrazeneca.com; lyndon.emsley@epfl.ch

between the target protein of interest (POI) and the E3-ligase which facilitates poly-ubiquitinoylation of the target and subsequent degradation by the proteasome. In order to form the ternary complex, PROTACs can be tuned to bind any ligandable domain in the POI (not necessarily the domain responsible for the disease), hence the growing interest in PROTACs today is motivated by the fact that they can in principle target POIs that were challenging until now.

Understanding the structure of PROTACs at the atomic level, and changes induced by binding to the two proteins is key to developing more efficient degraders in the future. For example, the effect of the linker structure on the improvement of the drug-like properties of the degrader has been emphasized in the past[12,13].

Amorphous solids are of high interest as pharmaceutical formulations, especially in the case of complex molecules such as PROTACS which in general do not readily crystallize. However, the risk of later spontaneous structural rearrangements due to the environment, that will change the bioavailability, prevents amorphous forms from being broadly applicable. This risk is difficult to determine a priori owing to a lack of experimentally determined atomic-level structures available. To better understand which structural features help stabilize amorphous formulations, chemically rich three-dimensional structures need to be determined. In this context, chemically rich means not only determining the conformation of a single molecule but also determining a structure that includes the surrounding molecules.

X-ray crystallography is extensively used to solve structures of crystalline molecular solids, as well as crystallized proteins and protein complexes. Typically for PROTACs, limited information on the structure of the drug is available from electron diffraction, especially cryo-EM[14–17], for the bound structure in the ternary complexes. Notably, neither X-ray diffraction nor cryo-EM can be applied to solve the atomic-level structure of amorphous PROTACs in their pure solid form, such that there is currently no knowledge of the amorphous structures.

In this context, chemical shift driven NMR crystallography can be used to determine atomic-level structures in solids[18,19], with applications extending from microcrystalline powders of small organic molecules[20–30], to enzyme active sites[31–33], to inorganic materials[34–37] and passivating layers in photovoltaic materials[38–40], or cements[41–46]. Structures are solved by comparing experimentally measured NMR chemical shifts with those calculated for candidate structures. The strength of this method is that NMR observables are sensitive to the local chemical environment and do not require long-range order for structure determination[18,47,48]. Indeed solid-state NMR has long been applied to disordered materials[18,49–55].

We have recently shown how complete atomic-level structures of amorphous molecular solids can be obtained by NMR crystallography[56–58]. The distribution of atomic-level environments present in a disordered sample leads to chemical shift distributions in the experimental spectra. While this leads to an added challenge for

spectral assignment because of spectral overlap, even for relatively small molecules[59,60], nevertheless, the signature of the complete atomic-level structure is directly contained within the chemical shift distributions from each nucleus.

To determine the complete structure from the chemical shift distributions, molecular dynamics is used to generate a large set of candidate structures, for which chemical shifts are predicted. This prediction step, which is key to establish the link between shifts and structure, has been enabled by the introduction of machine learning based chemical shift models[61–64], since traditional DFT methods would be prohibitively resource intensive for such large and numerous structures. Comparing the experimental shift distributions with the computed chemical shifts yields a subset of structures that are in best agreement with experiment (we will refer to this as the NMR set in the following). In this way, the complete amorphous structures of AZD4625 and Atuliflapon (AZD5718), including preferred conformations and H-bonding patterns, have been determined[56,65].

Here, we determine the atomic level structure of pure amorphous PROTAC 2 (Fig. 1). PROTAC 2 was developed via structure-based design to bind the BAF ATPase subunits SMARCA2 and SMARCA4 and the E3 ubiquitin ligase VHL[12,66]. In structure of amorphous PROTAC 2 we find significantly different levels of disorder for the three different functional parts of the molecule. The least disorder is seen in the SMARCA binding region, due to a strong intra-molecular H-bond. There is more disorder in the peptide-like VHL binding region, and the linker is almost completely disordered.

## Results and discussion
### NMR assignment
The assignment of the solid-state NMR chemical shifts of amorphous PROTAC 2 was obtained using $^1$H, $^{13}$C and $^{15}$N detected assignment strategies[67]. The assignment of the amorphous form is particularly challenging due to the relatively large size of the molecule (with 48 carbons and 35 inequivalent protons) which leads to significant spectral overlap in the solid-state MAS spectra. This is compounded by the lack of a crystalline form which would otherwise guide the assignment of the amorphous form, as in for example the case of Atuliflapon[57]. We therefore started by assigning the solution-state chemical shifts of PROTAC 2 dissolved in hexadeuterated DMSO (DMSO-d$_6$), as detailed in the SI. Figure S1 shows the 1D $^1$H spectra of the solution of PROTAC 2 acquired at variable sample temperatures in the range from 293 K to 318 K. The largest change in $^1$H chemical shifts over this temperature range is less than 0.05 ppm, which reduces the possibility of an erroneous solid-state assignment due to differences in sample temperature between the solution- and solid-state. The solution-state assignments (Table S4) were then compared with the peak positions in the one- and two-dimensional spectra acquired on the as received powder, and with the peak positions observed in a frozen solution of PROTAC 2. The spectra used in this comparison include short-range hCH, $^1$H-$^{13}$C DNP enhanced CP, $^{19}$F-$^{13}$C CP, DNP enhanced Cq-edited $^{13}$C spectra and the DNP enhanced attached $^{14}$N test. Acquisition parameters are given in the experimental section above, and in SI. Dynamic nuclear polarization (DNP) was used to enhance sensitivity for $^{13}$C and $^{15}$N detected experiments, especially in the experiments that suffer from fast relaxation throughout the pulse sequence, e.g., the attached $^{14}$N test, or experiments that rely on correlations between nuclei with low natural abundance, e.g., $^1$H-$^{15}$N HETCOR.

Assuming that the change in the state of the sample affects the width of the peaks to a much larger extent than their position in the spectrum[68], the assigned solution-state shifts were refined to match the nearest solid-state peaks. In this way, 41 carbon and 31 proton solid-state chemical shifts (i.e., all those assigned in solution except C23, C24, C36, C40, C46, C47, H24, H36, H40, H42, and H47) were unambiguously assigned (Table S5). These assignments are also in line with predictions from the assignments predicted using probabilistic

**Fig. 1 | Two-dimensional chemical structure of PROTAC 2.** The SMARCA2 binding region of the molecule, VHL binding region and the linker are represented in blue, red and black, respectively.

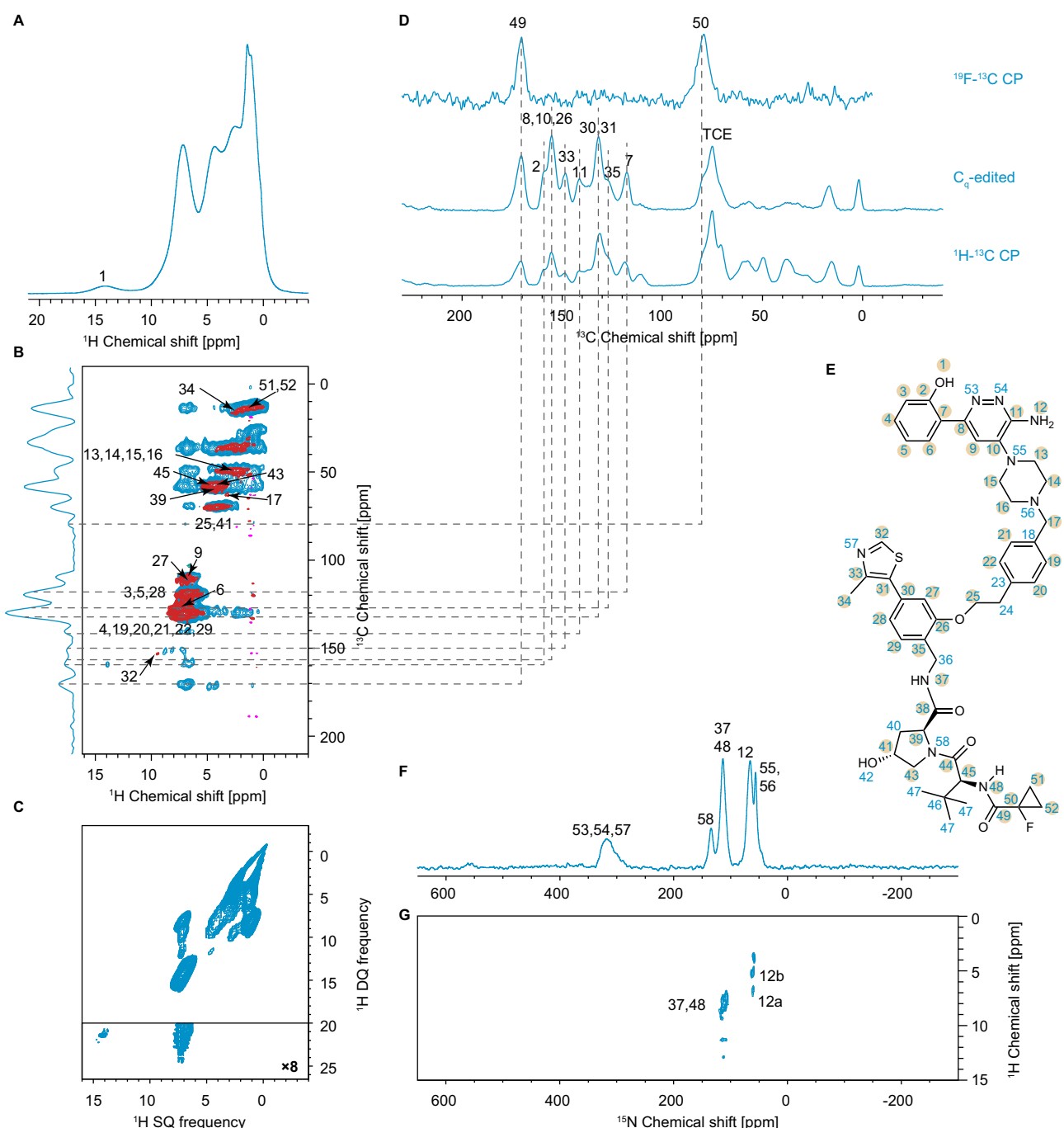

**Fig. 2 | Experimental NMR data for amorphous PROTAC 2. A** 900 MHz 1D $^1$H spectrum of the as-received powder of PROTAC 2 acquired at 293 K with 100 kHz MAS. **B** 900 MHz 2D hCH spectra acquired at 293 K and 100 kHz MAS and with a back-CP contact time of 5 ms and 250 μs in the blue and red spectra, respectively. The assigned correlations are labeled with the corresponding atomic sites. **C** 900 MHz 2D $^1$H-$^1$H BABA DQ/SQ spectrum acquired at 293 K and 100 kHz MAS. **D** 101 MHz $^1$H-$^{13}$C DNP enhanced CP, $C_q$-edited spectrum of the frozen solution of PROTAC 2 acquired at 100 K and 10 kHz MAS, and a $^{19}$F-$^{13}$C CP spectrum of the as-

received powder at room temperature and 11 kHz MAS. The peak at 75 ppm corresponds to the residual solvent tetrachloroethane (TCE). **E** Chemical structure of PROTAC 2 with the labelling scheme used here. The labels of atoms whose experimental chemical shift distributions are used for comparison with predicted shifts are highlighted by orange circles. **F** 40 MHz $^1$H-$^{15}$N DNP enhanced CP spectrum of the frozen solution of PROTAC 2, acquired at 100 K and 10 kHz MAS. **G** 400 MHz 2D $^1$H-$^{15}$N DNP enhanced DUMBO-HETCOR spectrum of the impregnated powder of PROTAC 2 acquired at 100 K and 10 kHz MAS.

methods[69], as shown in Fig. S4. The maximal absolute variation of the $^{13}$C chemical shifts between solution- and solid-state in this group is 3.0 ppm, corresponding to C14 and C16, as shown in Fig. S5. As shown in Fig. 2, the 1D $^{13}$C spectrum obtained from the projection of the hCH long-range spectrum acquired at room temperature shows peaks at similar positions as the $^1$H-$^{13}$C CP spectrum acquired at 100 K. We therefore assume that there is no significant large amplitude dynamics

present in the powder at room temperature that would otherwise need to be taken into account for the subsequent analysis.

For each atomic site, the experimental shift distribution was then determined by fitting the peak from a given spectrum to a Gaussian distribution. All the protonated carbons and their attached protons (except C13, C14, C15, C16 and their protons) were fitted using the extracted hCH short-range columns and rows, respectively (Figs. S6,

S9). All the non-protonated carbons except C18 and C23 were fitted using the $C_q$-edited spectrum (Fig. S7). The $^1$H-$^{13}$C CP spectrum was used to fit C13, C14, C15, C16 together to a single Gaussian peak and C18, C23 to another Gaussian distribution (Fig. S8).

The acidic proton H1 was assigned to 13.9 ppm based on the long-range hCH spectrum where it showed a correlation with C2. Since H1 is the only atomic site whose chemical shift corresponds to the peak at 13.9 ppm in the 1D $^1$H spectrum (with much higher sensitivity than the long-range hCH extracted row at the C2 chemical shift in F1), H1 was fitted directly from the 1D $^1$H spectrum at 100 kHz MAS and room temperature (Fig. S10).

The amide and amine protons H12, H37 and H48 were assigned and fitted using the $^1$H-$^{15}$N HETCOR spectrum (Fig. S11).

All the resulting mean average shifts μ and widths σ of the distributions are given in Table S5. The Mathematica notebooks used for solution-state assignment, solid-state assignment and solid-state fitting are available in the SI.

The atomic sites with measured distributions, that we use in the subsequent analysis, are shown on the molecule in Fig. 2. We note that the $^{13}$C linewidths are on the order of 1–3 ppm and broadening due to incomplete removal of the anisotropic interaction ($T_2'$)[70], and to magnetic susceptibility broadening[71] are within the error of the analysis (~50 Hz). Importantly, we note that for NMR-based structure determination, complete assignment of all the atomic sites in a molecule is not necessary, as the chemical shift of a given atomic site encodes information about the surrounding environment[56,58].

## Selection of best-matching structures

To characterize the structure of the amorphous solid, a set of structures is identified that are energetically reasonable and in best agreement with experimental data. Specifically, we use molecular dynamics to generate a diverse space of accessible chemical environments, resulting in 372,864 structures (as described above). To rank the agreement of these environments with the experimental data, the chemical shifts of all the atoms in each MD frame are predicted using a machine-learning model, ShiftML2[61]. The predicted shifts for all the environments in all the frames are then ranked using a p-value based on their probability of being within the experimental distribution. The p-value for a given environment is the geometric mean of the probability of the individual shifts, as described in ref. 57.

The experimentally determined set of structures is then composed of the 5,000 structures in best agreement with the experimental data. The lowest resulting p-value is 0.21. This NMR set is then analyzed and compared with all the structures generated from molecular dynamics, referred to as the MD set, to determine preferred interactions and conformations.

## Structural motifs in the experimental structure

First, we address hydrogen bonding. Hydrogen bonding has been shown to play a critical role in amorphous AZD4625[65] and Atuliflapon[49,56], where it was postulated that some conformations are promoted to enable more beneficial hydrogen bonding interactions. Figure 3a shows the hydrogen bonding motifs present in the NMR set and compared to the MD set. In the case of PROTAC 2, we observe a strongly promoted intramolecular interaction between H1 and N53, which is present in 80 % of the molecules in the NMR set. To further characterize this interaction, we compare the dihedral angle defined by C2-C7-C8-C9 and the H1-N53 distance in the NMR set with those in the MD set in Fig. 3c, d. For the C2-C7-C8-C9 dihedral angle, the distributions near 180° are strongly promoted, while those centered near 0° are demoted in the NMR set. Figure 3e shows how the dihedral angle is very strongly correlated to the H1-N53 distance which is a reporter of the H-bonding interaction.

The promotions of the dihedral angle in the NMR set support the hypothesis that low-energy conformational changes occur to promote

more strongly stabilizing interactions. In this case, this corresponds to the whole phenol ring moiety being locked into position to maintain the hydrogen bond between H1 and N53. Figure 4 shows a superposition of 100 structures with the H1-N53 H-bond present from the NMR set for this fragment of the molecule. The figure clearly shows how when the H1-N53 H-bond is present, it locks the rest of this phenol-pyridazine fragment of the molecule into a quite highly ordered structure.

It should be noted from Fig. 3a that for 20 % of the molecules in the NMR set, H1 has other intermolecular interactions (i.e., with amines and carbonyls) or no H-bonding interactions (2%). However, all these motifs are demoted in the NMR set as compared to the MD set.

We see from the calculated relative cluster formation energies, shown in Fig. 3b, that the promoted interaction (H1-N53) provides ~10 kJ/mol of stabilization as compared to motifs with no hydrogen bond. Interestingly, though, it is not predicted to be the most stabilizing H-bonding interaction, with, for example, motifs having an intermolecular H1-O49 hydrogen bond yielding ~30 kJ/mol stabilization as compared to no hydrogen bond (Fig. 3b). However, these other potentially highly stabilizing motifs are not found to present at significant levels in the amorphous structure.

We also analyzed the other hydrogen bond interactions for H12a, H12b, H37, and H48 shown in Figs. S12–S14. (Note that H42 was also examined but the chemical shift could not be assigned). We do not observe any strong promotion of any hydrogen bond acceptors for these protons.

Thus, we conclude that in contrast to AZD4625 and Atuliflapon, where intramolecular H-bonds were found to be an important contribution to stabilization of the solid form, here we postulate the main stabilizing interaction for the amorphous solid form of PROTAC 2 is not hydrogen bonding.

Now we turn to the conformation of the piperazine ring. For AZD4625 and Atuliflapon, the amorphous structures both showed promoted conformations in the six-membered aliphatic rings. PROTAC 2 also contains a piperazine ring and we use the dihedral angle N55-C13-C14-N56 to report on the conformations observed in the experimental structure. Figure S15 shows the populations seen in the NMR and MD sets, where we observe a minor promotion in the distribution of angles centered at about ±60 degrees. Strikingly, we observe the energy of the NMR set is lower for dihedral angles centered at ±60 degrees as compared to the MD set despite energy playing no direct role in the structure selection. Notably, in conclusion, both of the major conformers are preset in the NMR set, with some disorder.

Next, we examine the structure of the peptide-like region. PROTAC 2 contains a five-membered aliphatic ring that is connected within the peptide-like motif of the molecule. As above, we examine the ring conformation using two dihedrals, C44-N58-C39-C38 and C40-C39-N58-C43. In Fig. 5, both dihedrals show a broad distribution of angles centered at ~−75 and ~−20 degrees, respectively, indicating a large degree of disorder, with no significant difference between the MD and the NMR sets. For both angles, there is a clear energy minimum at ~−100° and ~−30°, respectively.

We then look at the backbone structure of the molecule here and contrast this ring against the expectations for a similar peptide structure, proline. In proline the backbone angles φ and ψ are distributed around −65 degrees and −35 and 150 degrees, respectively, in protein structures[72,73]. For amorphous PROTAC 2, we see in Fig. 6a, that φ is observed to have a single distribution centered at ~−75 degrees and for ψ we observe a broad distribution with three peaks centered about −35, 50, and 150 degrees. There is a slight promotion in the NMR set for the ψ angle centered at 50 degrees, and a demotion for the angles at 150 degrees. Interestingly, the demoted conformation is typically observed in β-regions and is the conformation that is found when PROTAC 2 is bound to SMARCA and VHL[66] (PDB: 6HAX).

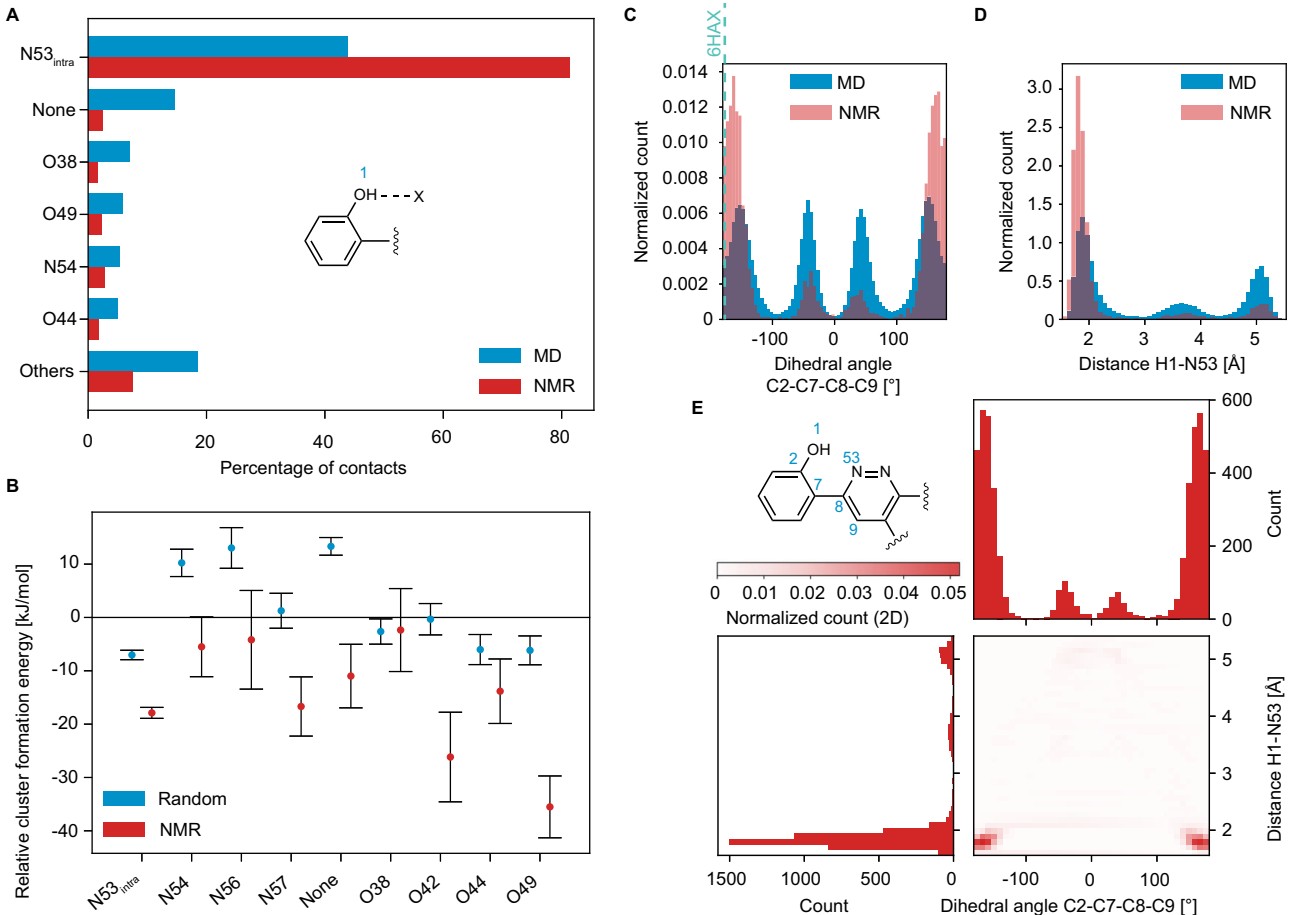

**Fig. 3 | Hydrogen bonding. A** Occurrence of different H-bond acceptors for H1 in the MD set (blue) and NMR set (red). **B** Relative formation energies of the local molecular environments in 12,000 structures randomly selected from the MD set (blue) and the 5000 structures contained in the NMR set (red) for different H-bond acceptors bonded to H1. The error bars represent one standard deviation of the distribution of the mean relative formation energy across all environments within the same bin of the geometric feature. **C** The histogram of the dihedral angle C2-C7-C8-C9 in the MD set (blue) and the NMR set (red). The vertical teal dashed line corresponds to the dihedral angle value of −177° found when PROTAC 2 is bound to SMARCA and VHL[66] (PDB: 6HAX). **D** The histogram of the intramolecular distance H1-N53 in the MD set (blue) and the NMR set (red). **E** 2D correlation plot between the histogram of the dihedral angle C2-C7-C8-C9 and the histogram of the distance H1-N53 in the NMR set.

Furthermore, in Fig. S15, we show the distributions for dihedral angles between the carbonyl orientations in the peptide region, (O38-C38-C44-O44 and O44-C44-C49-O49). For both angles, the distribution is very broad, indicating a large degree of disorder.

However, we do note the strong correlation seen in Fig. 6b between the "proline" ψ angle and the dihedral angle between O38-C38-C44-O44. So, while this region of the molecule is overall highly disordered, there are structural correlations within individual environments.

It is interesting to note that this dihedral angle is −102 degrees in the protein complex (as shown in Fig. 6b), which is notably infrequent in the NMR set. We postulate differences such as this between the amorphous solid formulation and PROTAC 2 in complex with SMARCA and VHL are a result of the hydrogen bonds formed when PROTAC 2 is bound. In the complex, O38 is 2.8 Å from $O^{\eta}$ of Tyr98 and O44 is 2.7 Å from the oxygen of a crystallographic water[66].

Finally, we evaluate the degree of long-range order in the structure. PROTAC 2 contains a flexible linker between the two ligands, which can contribute to structural disorder within the molecule. To understand the overall disorder present in the amorphous solid form, we examine the dihedral angle between C20-C23-C26-C27 (the dihedral angle across the linker) shown in Fig. S15 and the distances between C7-C26, C7-C50, and C18-C39 shown in Fig. S16. For the dihedral angle across the linker, we see from Fig. S15 that virtually all angles are possible, that there is no promotion or demotion of any

distribution of angles in the NMR set, and we see and there is no energetic advantage for any particular dihedral angle. This is expected as the linker is a flexible aliphatic chain with four degrees of freedom and no strong hydrogen bond acceptors or donors. Similarly, for each of the measured distances, we observe a continuous distribution spanning -12.5 Å, -22.5 Å, and -12.5 Å for C7-C26, C7-C50, and C18-C39, respectively. As before, there is no promotion for any given distance. However, we do see a clear energetic minimum for the more extended distances between atomic sites.

Interestingly, as shown in Fig. 7a, there is a correlation between the C7-C50 distance and the dihedral angle between N56-C17-C18-C19. This correlation is seen specifically when the dihedral angle is in the distribution centered about -90 degrees, with the C7-C50 distances between 10–15 Å. Additionally, we see the greatest difference between the energy of the MD set and the NMR set for distances between 10–15 Å.

In summary, the linker section of the molecule is highly disordered, in a way that is reminiscent of glassy polymers. We note in passing that the Tg of the sample (-105 °C) is similar to polymers such as polystyrene (-100 °C) and co-povidone (-110 °C), which also have mixed rigid and flexible parts of their structures.

In conclusion, we have determined the complete atomic-level structure for amorphous active pharmaceutical ingredient (API), PROTAC 2, using chemical shift driven NMR crystallography. Overall, the structure of amorphous PROTAC 2 is much more

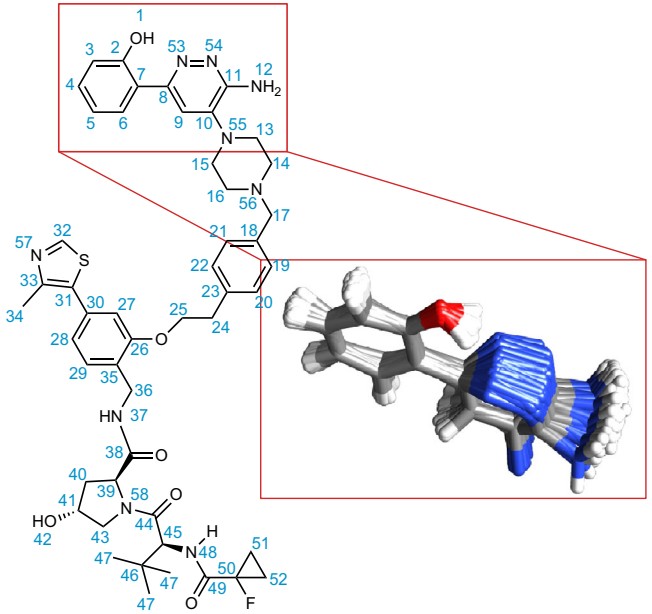

**Fig. 4 | Intramolecular hydrogen bonding between H1 and N53.** Superposition of 100 molecules of PROTAC 2 randomly selected from a subset of the NMR set lying within the promoted regions of the dihedral angle H1-O1-C2-C7 between −30° and 30° and the dihedral angle C2-C7-C8-C9 less than −150° or greater than 150°, and then aligned using carbon atoms in the phenolic ring, as well as C8 and N53.

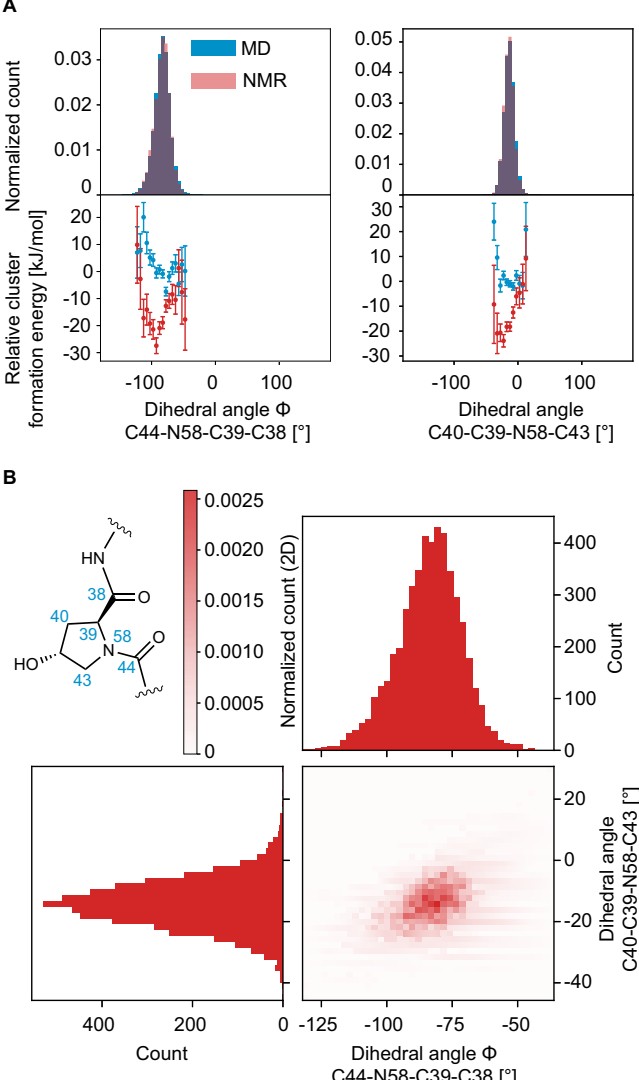

**Fig. 5 | Structural features of the peptide-like region. A** On the left, the histogram of the dihedral angle C44-N58-C39-C38 in the MD set (blue) and the NMR set (red) and the relative formation energies of the local molecular environments in 12,000 structures randomly selected from the MD set (blue) and the 5000 structures contained in the NMR set (red). On the right, the histogram of the dihedral angle C40-C39-N58-C43 in the MD set (blue) and the NMR set (red) and the relative formation energies of the local molecular environments in 12,000 structures randomly selected from the MD set (blue) and the 5,000 structures contained in the NMR set (red). The error bars represent one standard deviation of the distribution of the mean relative formation energy across all environments within the same bin of the geometric feature. **B** 2D correlation plot the histogram of dihedral angle C44-N58-C39-C38 and the histogram of the dihedral angle C40-C39-N58-C43 in the NMR set.

conformationally disordered as compared to the structures of the amorphous forms of the smaller compounds AZD4625 and Atuliflapon. Also, by comparison, AZD4625 consists mainly of fused rings, with significant steric hindrance, leading to the promotion of specific conformations in the experimental structure. In contrast, PROTAC 2 has a linker with an aliphatic chain that allows for more polymer-like solid-state structures.

We note that in Atuliflapon, there are several promoted hydrogen bonding interactions with the NH group, which may drive conformations required to form this stabilizing interaction. In the case of PROTAC 2, we see only a single promoted intramolecular hydrogen bond between H1 and N53. While this interaction drives the promotion of conformations of the phenol ring, it has little impact on the larger structural order as noted by the relatively small difference between the energy for the same interaction for the NMR and MD sets.

We find that the conformations in the peptide-like VHL binding region are correlated with the ψ and φ of the "proline" showing order on the local scale. However, this dihedral angle covers a large space even for the promoted conformations (0–100 degrees), suggesting no long-range order on the bulk scale.

Overall, in PROTAC 2, we find the least disorder in the SMARCA binding region, with relatively well-defined ring structures locked into place by a strong intra-molecular H-bond. There is more disorder in the peptide-like VHL binding region, and the linker is almost completely disordered. Thus, in contrast to the previously studied structures of smaller molecular drugs, where intermolecular H-bonding interactions were found to be the main stabilization mechanism for the amorphous solid form, in PROTAC 2 we postulate that, in analogy to glassy polymers, the main stabilization mechanism of the amorphous form is the entropic contribution introduced by the overall flexibility, especially in the linker region of the molecule.

## Methods

PROTAC 2 was synthetized according to literature procedure with a purity above 90% and supported by x-ray diffraction. The experimental procedure for the synthesis of PROTAC2 as well as the characterization

data for key intermediates are given in the SI. An amorphous powder hereof was obtained by concentrating PROTAC 2 in vacuo from a mixture of water and acetonitrile[74,75]. Thermal characterization of the powder by DSC shows the glass transition temperature is ~105 °C. Solid-state NMR measurements were performed using the amorphous solid as (i) a dry powder, (ii) an impregnated powder, or (iii) a frozen solution (as detailed below). The as-received powder was crushed with mortar and pestle before packing inside a 0.7 mm rotor.

For dynamic nuclear polarization (DNP) measurements, continuous wave microwave irradiation was applied where the frequency was optimized using a field-sweep coil so as to give maximum DNP enhancement. The optimal microwave output power of the gyrotron

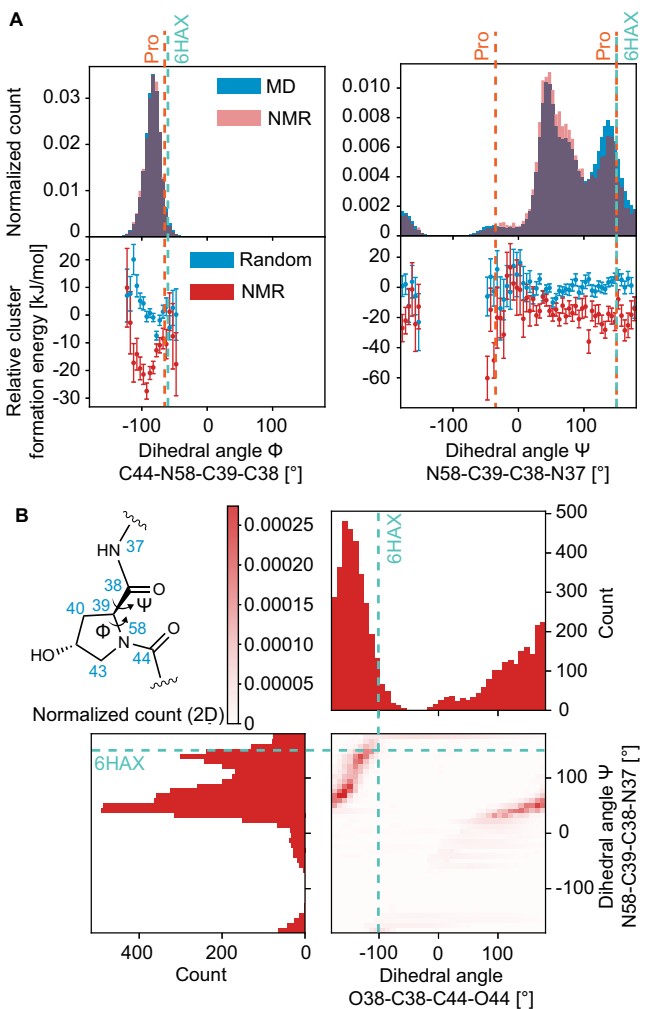

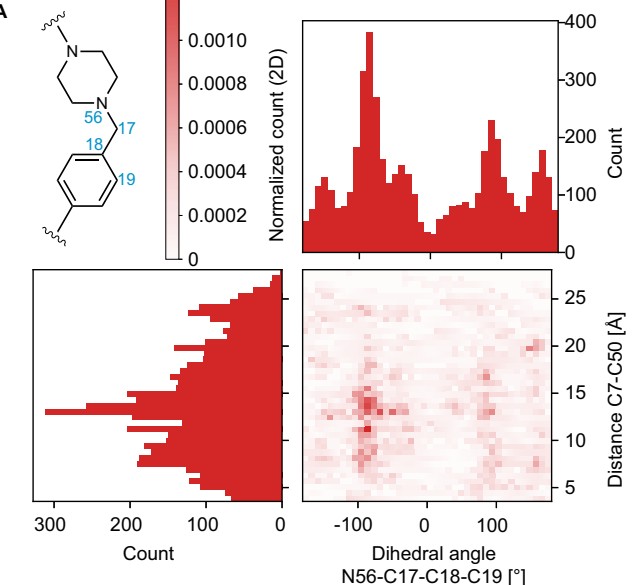

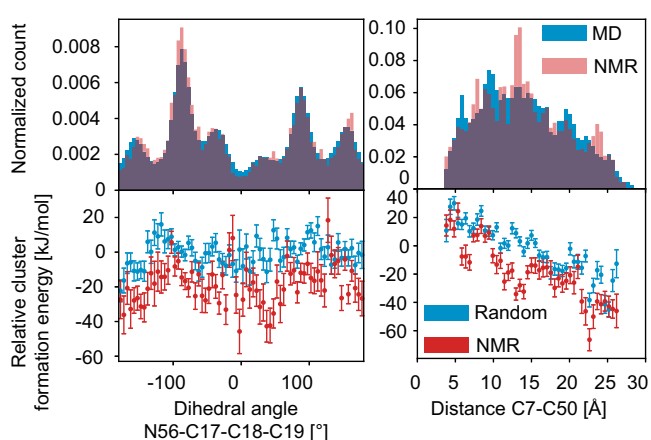

**Fig. 6 | Distributions of dihedral angles around the proline group in the peptide-like region. A** On the left, the histogram of the dihedral angle C44-N58-C39-C38 in the MD set (blue) and the NMR set (red) and the relative formation energies of the local molecular environments in 12,000 structures randomly selected from the MD set (blue) and the 5,000 structures contained in the NMR set (red). The vertical teal dashed line corresponds to the dihedral angle value of −60° found when PROTAC 2 is bound to SMARCA and VHL[66] (PDB: 6HAX). The vertical orange dashed line corresponds to the dihedral angle value of −65° found in proline in protein structures. On the right, the histogram of the dihedral angle N58-C39-C38-N37 in the MD set (blue) and the NMR set (red) and the relative formation energies. The vertical teal dashed line corresponds to the dihedral angle value of 150° found when PROTAC 2 is bound to SMARCA and VHL[66] (PDB: 6HAX). The vertical orange dashed lines corresponds to the dihedral angle values of −35° and 150° found in proline in protein structures. The error bars represent one standard deviation of the distribution of the mean relative formation energy across all environments within the same bin of the geometric feature. **B** 2D correlation plot the histogram of dihedral angle O38-C38-C44-O44 and the histogram of the dihedral angle N58-C39-C38-N37 in the NMR set. The teal dashed lines corresponds to the dihedral angle values of 150° and −102° for N58-C39-C38-N37 and O38-C38-C44-O44, respectively, found when PROTAC 2 is bound to SMARCA and VHL[66] (PDB: 6HAX).

**Fig. 7 | Long-range order. A** 2D correlation plot the histogram of dihedral angle N56-C17-C18-C19 and the histogram of the distance C7-C50 in the NMR set. **B** On the left, the histogram of the dihedral angle N56-C17-C18-C19 in the MD set (blue) and the NMR set (red) and the relative formation energy of the local molecular environments in 12,000 structures randomly selected from the MD set (blue) and the 5000 structures contained in the NMR set (red) as a function of the dihedral angle. On the right, the histogram of the distance C7-C50 in the MD set (blue) and the NMR set (red) and the relative formation energies of the local molecular environments in 12,000 structures randomly selected from the MD set (blue) and the 5000 structures contained in the NMR set (red) as a function of the distance. The error bars represent one standard deviation of the distribution of the mean relative formation energy across all environments within the same bin of the geometric feature.

sources was found at a cathode voltage of 16.9 V and collector current between 140 and 160 mA.

15.1 mg of PROTAC 2 crushed with a mortar and a pestle was impregnated with 7.5 uL of a 12 mM solution of AMUPol[76] in glycerol-$d_8$:D$_2$O:H$_2$O 60:30:10 v/v. The impregnated powder was packed in a 3.2 mm sapphire rotor. Upon continuous irradiation of the sample with microwaves, the signal-to-noise ratio per square root of the experimental time was 375/√h.

The frozen solution was composed of PROTAC 2 dissolved in a solution of 16 mM TEKPol[77] in tetrachloroethane (TCE) resulting in a 0.51 M solution of PROTAC 2. The frozen solution was packed in a 3.2 mm sapphire rotor and subjected to 3−5 freeze-thaw cycles inside the probe to minimize the amount of dissolved oxygen[78]. Upon continuous application of microwaves, the signal-to-noise ratio per square root of experimental time was 271/√h.

## NMR experiments

All spectra of the dry powder, except the $^{19}F$ to $^{13}C$ cross-polarization (CP) spectrum, were acquired on a Bruker Avance Neo spectrometer operating at 21.14 T (900 MHz and 225 MHz $^1H$ and $^{13}C$ frequencies,

respectively) equipped with a 0.7 mm room temperature HCN CP magic angle spinning (MAS) probe. The spectra were acquired at 100 kHz MAS at a sample temperature of 293 K, except where stated otherwise. The one-dimensional $^1H$ spectrum was recorded using a rotor-synchronized spin echo with the echo delay equal to three rotor periods for background suppression and a recycle delay of 2.1 s. The 2D short-range hCH NMR spectrum[79] was recorded with 500 µs and 250 µs contact times, for the direct $^1H$-$^{13}C$ CP and for the $^{13}C$-$^1H$ back-CP, respectively. The long-range 2D hCH was recorded with 5 ms and 5 ms contact times for the direct $^1H$-$^{13}C$ CP and for the $^{13}C$-$^1H$ back-CP, respectively. WALTZ-16[80] decoupling at $^1H$ and $^{13}C$ nutation frequency of 10 kHz was applied during $t_1$ and $t_2$, respectively. In total, 48 increments with 2048 scans each for long-range hCH, and 128 increments with 1024 scans each for the short-range hCH, each with a recycle delay of 2.1 s, were acquired, resulting in 480 µs and 1.4 ms, respectively, of total evolution in the indirect dimension of the long- and short-range hCH. In the $^1H$ 2D DQ/SQ BABA-xy16[81] spectrum, one DQ excitation and reconversion period of eight rotor periods each, was used with a total of 400 increments with 16 scans each were acquired with a repetition delay of 2.1 s. The acquisition time was 9 ms and 4 ms in $t_2$ and $t_1$, respectively.

In addition, a series of one-dimensional $^1H$ spectra were recorded at a sample temperature of 259 K to 329 K. For that, a rotor-synchronized spin echo with the echo delay equal to one rotor period for background suppression and a recycle delay of 1 s were used.

The one-dimensional $^{19}F$ to $^{13}C$ CP spectrum of the dry powder was acquired at room temperature and 11 kHz MAS in 8192 scans using a 90% to 100% ramp on $^{19}F$ channel during the CP step and a 3000 ms contact time.

The DNP experiments on the frozen solution were carried out on a commercial Bruker Avance III spectrometer operating at 9.39 T ($^{13}C$ Larmor frequency of 101 MHz), equipped with 3.2 mm H/X/Y LTMAS probes connected through a corrugated waveguide to a 264 GHz klystron[82]. The temperature was set to 100 K and the MAS rate was 10 kHz. The recycle delay was set to 1.3 times the $^1H$ build-up time for optimal sensitivity under DNP.

In all the DNP enhanced one-dimensional $^{13}C$-detected experiments, the $^1H$ to $^{13}C$ CP contact time was set to 2 ms with a 90% to 100% ramp on $^1H$ channel. The $^1H$ to $^{13}C$ CP spectrum was acquired in 32 scans. The $^{13}C$ spectrum with an echo period of 300 µs without $^1H$ decoupling to yield a spectrum consisting of resonances corresponding to quaternary carbons ($C_q$-edited) was acquired with 64 scans. The attached nitrogen test[83] via $^{13}C$-$^{14}N$ RESPDOR dephasing used to identify carbons directly bound to nitrogen in order to refine the assignment was acquired in 8192 scans.

The 1D $^1H$ to $^{15}N$ CP spectrum of the frozen solution was acquired at 10 kHz MAS in 4096 scans using a 70% to 100% ramp on $^1H$ channel during the CP step and a 5000 ms contact time.

The DNP enhanced 2D $^1H$-$^{15}N$ HETCOR[84,85] experiment with homonuclear dipolar decoupling in $t_1$ on the impregnated powder of PROTAC 2 was carried out on commercial Bruker Avance III spectrometer operating at 9.39 T ($^{13}C$ Larmor frequency of 101 MHz), equipped with 3.2 mm H/X/Y LTMAS probes connected through a corrugated waveguide to a 263 GHz gyrotron microwave source[82]. The spectrum was acquired at 10 kHz MAS and 100 K. An eDUMBO-1$_{22}$[86] homonuclear dipolar decoupling element (32 µs cycle time at $v_1$ = 91 kHz) was applied during $t_1$ to increase the resolution of the indirect dimension by reducing the contribution of the $^1H$-$^1H$ homonuclear dipolar couplings. The scaling factor in the indirect dimension was 0.676. A $^1H$-$^{15}N$ contact time of 1.5 ms and a 50% to 100% ramp on the $^1H$ channel was used during the CP step. SPINAL-64 decoupling[87] with an rf amplitude of 80 kHz was applied during the acquisition. 128 increments of 32 scans each were acquired.

States-TPPI was used for quadrature detection in the indirect dimension for all 2D experiments.

Proton chemical shifts were referenced externally with respect to the adamantane $CH_2$ group at 1.87 ppm. Carbon chemical shifts were referenced externally with respect to the adamantane CH group at 38.48 ppm[88]. $^{15}N$ chemical shifts were externally referenced with respect to $NH_4Cl$ at 39.3 ppm at room temperature[89].

All the detailed acquisition parameters and a link to the raw NMR data are available as described in SI (see Table S1).

## MD simulation

Molecular dynamics simulations were carried out for eight independent simulations. Each simulation consisted of a cell containing 128 PROTAC 2 molecules. For three of the simulations, the starting conformation of the amide bond(s) was set to cis. More details and a link to the raw data are given in SI.

## Chemical shift prediction

The snapshots obtained from molecular dynamics were used as input for ShiftML2[61] where chemical shifts were predicted for $^1H$, $^{13}C$, and $^{15}N$. To convert from shielding to shift, a slope of −1 and an offset of 30.78, 170.04, and 227.9 for $^1H$, $^{13}C$, and $^{15}N$ were used, respectively[57]. A link to the ShiftML output is given in SI.

## Calculation of formation energies

The formation energies of local molecular environments were computed as described in ref. [57]. The environments contain a central molecule and all surrounding molecules with at least one atom within 7 Å of the central molecule. Three energy calculations were performed for each cluster, for (i) the environment containing the central molecule, (ii) the environment without the central molecule, and (iii) the central molecule without the surrounding environment. The energies were computed using the DFTB-D3H5 semiempirical level of theory using the 3ob-3-1 parameter set and the DFTB+ software version 24[90–96]. The difference in energy between the environments with and without the central molecule provides an estimate of both the conformational energy of the central molecule and the energy of intermolecular interactions with the environment.

The energies of 5000 molecular environments that compose the NMR set were computed and compared to the energies of an ensemble of 12,000 randomly selected molecular environments from the MD simulations (1500 environments selected per MD run).

## X-ray characterization

X-ray diffraction was carried out in reflection geometry on a Bruker D8 Discover Plus system equipped with a Cu rotating anode and a Dectris Eiger2 500 K detector. The beam was shaped with a 60 mm focussing Göbel mirror, data was acquired at 1 s/step acquisition time and a step size of 0.3°, using the 1D mode of the detector at an opening of 2 × 2°. The measurement was carried out at room temperature from 3° to 80° scattering angle.

## Differential scanning calorimetry

Differential Scanning Calorimetry (DSC) experiments were carried with a DSC Discovery 2500 equipped with a Refrigerated Cooling System, in the temperature range from - 50 °C to 250 °C, at a modulated heating rate of 5 °C/min and modulation parameters of 1 °C for 60 s. Samples of approximately 2 mg were encapsulated in Tzero (aluminum) non-hermetic pans, to allow evaporation of water for determination of the dry glass transition. During all measurements, the calorimeter head of the DSC was flushed with highly pure nitrogen gas (flow rate of 50 mL/min). Temperature and enthalpy (cell constant) calibrations were based on the melting peak of the indium standard (Tm = 156.6 °C) supplied by TA Instruments, using the same experimental conditions (cell environment, type of pan/lid and heating rate) as for the samples.

## Supporting information

Experimental, fitting and computational details for the MD simulations, assignment details, additional NMR spectra and comparative histograms between the MD and the NMR ensembles, and a link to the raw NMR data, as well as solution NMR, HRMS and powder X-ray diffraction data for PROTAC 2.

## Data availability

The raw NMR data in JCAMP-DX version 6.0 standard format and the original TopSpin format, MD simulations, MD snapshots extracted, and formation energies of intermolecular complexes data used and/or generated in this study have been deposited in Materials cloud repository [https://doi.org/10.24435/materialscloud:d0-ck]. The data generated in this study are provided in the Supplementary Information/Source Data file. All data are available under the license CC-BY-4.0 (Creative Commons Attribution-ShareAlike 4.0 International). All data underlying this study are available from the corresponding authors upon request.

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

## Acknowledgements

This work was supported by AstraZeneca (L.E.), Swiss National Science Foundation Grant No. 200020_212046 (L.E.), and by the NCCR MARVEL (L.E.).

## Author contributions

Conceptualization: L.E., S.S., S.O.N.L., J.B.H., D.T., and E.M. Methodology: M.C., P.M., J.B.H., D.T., L.v.B., B.M., S.O.N.L., S.S., and L.E. Investigation: D.T., J.B.H., M.C., P.M., L.v.B., S.D.F., F.E., J.R.J., A.G.M., and E.M. Visualization: J.B.H., D.T., and L.E. Supervision: L.E., S.S., S.O.N.L., B.M., F.E. Writing (original draft): D.T., J.B.H. Writing (review and editing): D.T., J.B.H., M.C., P.M., L.v.B., F.E., E.M., S.D.F., J.R.J., A.G.M., S.O.N.L., B.M., S.S. and L.E.

## Competing interests

The authors declare no competing interests.
