## [Transparent Peer Review file · Nature Communications]

Determination of key functional structures of an amorphous VHL-based SMARCA2 PROTAC

Corresponding Author: Professor Lyndon Emsley

Version 0:

Reviewer comments:

Reviewer #1

(Remarks to the Author)

The authors have submitted a nice study of the characterization of the amorphous form of PROTAC 2. It is notoriously difficult to determine the structure of an amorphous molecule, especially a relatively large one, because of the distributions in conformational states that exist, compared to a crystalline form.

The primary focus of this paper is the structural characterization of the PROTAC 2 using solid-state NMR spectroscopy, complemented with molecular dynamic simulations, to fully understand the structure of the PROTAC 2 in the amorphous state.

Noteworthy results:

determining structures of amorphous molecules is challenging, and this represents a next level assignment of such a large molecule.

Significance to field and comparison to literature:

the ability to solve the structure of a molecule this complex in the solid state is something that I would not thought possible a few years ago. it takes a complex set of NMR experiments, including high-resolution proton NMR and sensitivity-enhanced DNP NMR to be able to get the information from the system. the ability to model these systems computationally is also quite helpful, but the combination is needed to extract all of the information needed

literature - the closest work that has been done in this area is references 51 and 54, which includes a previous nature communication on a much less complex system

Flaws:

Not really a flaw, but I do wish that the authors would comment on the potential conformational distributions that would exist between the material in the powder form vs that of the frozen solution. In particular, there is a potential for structural relaxation to occur in the powder state, resulting in a smaller range of conformations as the material eliminates the higher energy conformations. I would not anticipate seeing this in the frozen state, so it would be interesting to compare the carbon spectrum of the frozen solution to that of the powder and look for any differences.

Methodology:

this is work coming from the leading group in solid-state NMR spectroscopy, and clearly exceeds the standards in the field

Reproducibility:

the work can be easily reproduced, provided that the instrumentation and skills to acquire and interpret the NMR data were available to do so - that being said, there are very few places in the world where this work could be done

Overall comments:

This is an exquisite study of the amorphous structure of a relatively large molecule. The only part of the manuscript that is not of high quality is the introduction. I found it to be poorly written and not motivating the problem particularly well. For example, the sentence "This leads to high interest in the atomic-level structure of amorphous forms, since, in terms of bioavailability, the risk of later structural changes due to environmental factors prevents them from being broadly applicable." I have no idea what the authors are trying to say - is it that the material might crystallize? What structural changes would lead to changes in bioavailability, short of crystallinity? The compound does not appear to crystallize.

Minor comments:

P.2 it is labeled Scheme 1, but should probably be Figure 1

Reviewer #2

(Remarks to the Author)

Reviewer #3

(Remarks to the Author)

This is a very nicely described and clearly evidenced NMR crystallography study of a large molecule that exhibits both short and long range disorder. The molecular disorder makes this a challenging case for structure elucidation, and the work clearly describes the processes employed. The conclusions are well evidenced through those descriptions.

The authors could comment on the extent to which the ^{13}C and ^{15}N solid-state NMR line shape dependence on conformational distribution v T2 effects and line broadening due to incomplete averaging of X-1H dipolar coupling affects their analysis and resulting structural parameters they discuss. This is a common issue for NMR in structure determination and this paper could become a point of reference to highlight when the contributions to NMR line shape from factors other than molecular conformation distribution can be ignored and how to test whether that is the case.

Otherwise the paper is an exemplary study, excellent compilation of NMR parameters, assignment etc in the SI, and methods in the main text, certainly sufficient to be reproduced.

Reviewer #4

(Remarks to the Author)

Emsley, Schantz, and coworkers submitted a manuscript reporting the structural determination of an amorphous drug molecule, PROTAC 2. Amorphous solid dosage forms represent a major category in modern pharmaceuticals, yet their molecular-level structural understanding remains highly limited. Conventionally, drug molecular structures are determined in crystalline form by single-crystal X-ray diffraction or in organic solvents using solution NMR, primarily assisting discovery and process chemists during early drug development stages. However, most commercial drug products are formulated in amorphous or partially amorphous forms, and their structures are not meaningfully elucidated in these solid states. This presents a significant knowledge gap in understanding their physical stability, potential instabilities, and functional properties relevant to drug delivery.

Structural determination of amorphous materials remains a pioneering research area, with only a limited number of examples reported to date, particularly in pharmaceutical sciences. This study carries high novelty by utilizing a combination of advanced solid-state NMR techniques (including ultra-fast MAS, DNP enhancement), machine-learning-based chemical shift prediction, and molecular dynamics simulations to address a challenging and important problem. The experimental design and data quality are very strong, even considering the expected broad lines typical for amorphous APIs.

I highly recommend this manuscript for publication. The work is novel, rigorous, and timely. I only provide a few questions and comments below for the authors to consider during revision.

(1) The authors highlight the role of an intramolecular hydrogen bond (H1-N53), which provides stabilizing energy of approximately 10 kJ/mol. However, compared to potentially stronger intermolecular hydrogen bonding, this stabilization appears modest. Could the authors elaborate on why these stronger intermolecular interactions are not significantly populated? Is this due to kinetic trapping, steric constraints, or entropic penalties associated with organizing multiple chains? As a relevant reference, in a previous study on posaconazole [Lu et al., J. Phys. Chem. B, 2020, 124 (25), 5271-5283], intermolecular hydrogen bonding was shown to play a critical role in stabilizing amorphous drug structures via locally ordered domains. could the authors comment whether similar local ordering contributes meaningfully to PROTAC 2 stabilization, or whether entropic flexibility dominates?

(2) The analogy made to polymer-like amorphous behavior is very insightful. It would further strengthen the manuscript if the authors could correlate this observation to known polymer glass transitions and conformational entropy contributions, especially in relation to the glass transition temperature of PROTAC 2 and the high flexibility of its linker region.

(3) There are a few carbons that remain unassigned. Some of these, such as C34 and C36, exhibit characteristic chemical shifts (~35.2 and 37.7 ppm, respectively) from solution NMR data and are likely located in the linker region. Could the authors elaborate on the reasons why these peaks could not be assigned in the solid-state spectra? Since chemical shift assignment is critical for downstream structure determination, it would strengthen the manuscript if the authors could attempt to complete assignments for the remaining resolved peaks, particularly those visible in Figure 2. Furthermore, the authors are encouraged to illustrate the assignments more explicitly by including expanded or annotated spectral plots to complement Figure 2.

Minor comments and suggestions:

(4) The title could be slightly more precise in emphasizing the methodology, for example:
"Solid-State NMR Structural Determination of an Amorphous VHL-based SMARCA2 PROTAC."

(5) In the MD simulation section, please clarify whether cis-amide conformations contribute significantly to the final NMR-selected ensemble.

(6) As a suggestion, the authors may consider adding a summary table listing the total number of ¹H, ¹³C, and ¹⁵N sites in PROTAC 2 and how many have been assigned in both solution-state and solid-state NMR. This would help readers appreciate the extent of assignments.

(7) In the Introduction section, the authors have included many valuable, but largely non-pharmaceutical, references on the use of ssNMR in disordered materials, along with a few pharmaceutical examples that primarily originate from their own research group. Despite the limited amount of existing literature directly probing the structure and interactions of amorphous pharmaceuticals, several highly relevant studies appear to be missing. For example: *Molecular Pharmaceutics*, 2016, 13 (11), 3964-3975; *Molecular Pharmaceutics*, 2019, 16 (6), 2579-2589; *Molecular Pharmaceutics*, 2020, 17 (7), 2585-2598; *Journal of Pharmaceutical and Biomedical Analysis*, 2019, 165, 47-55; *Molecular Pharmaceutics*, 2020, 17 (6), 2196-2207; *The Journal of Physical Chemistry C*, 2022, 126 (29), 12025-12037.

Version 1:

Reviewer comments:

Reviewer #1

(Remarks to the Author)

We thank the authors for addressing the comments that were made previously. The only additional comments that I would make would be to recommend that the paragraph in the introduction beginning "Amorphous solids ..." be cut. The revision does not really address the fundamental pharmaceutical concerns, and the manuscript stands alone just based upon the need to understand the structure of complex amorphous systems.

As a minor comment, with the addition of the sentence:

"We note in passing that the T_g of the sample (~105 °C) is similar to polymers such as polystyrene (~100 °C) and copovidone (~110 °C), which also have mixed rigid and flexible parts of their structures."

it should be noted that only higher molecular weight (30 - 1,000 K) polystyrene has a glass transition temperature around 100 C. Lower molecular weights, e.g. 1K (~40-60C) and 5K (~70-80C), have a much lower glass transition temperature. Just adding "high molecular weight" in front of polystyrene would solve this. Copovidone does not really have different molecular weights in practical use, so no need to add a comment about molecular weight.

Reviewer #2

(Remarks to the Author)

Reviewer #4

(Remarks to the Author)

The authors have adequately addressed my questions and comments and have updated the manuscript accordingly. In my opinion, it is now ready for publication.

REVIEWER COMMENTS

Reviewer #1 (Remarks to the Author):

The authors have submitted a nice study of the characterization of the amorphous form of PROTAC 2. It is notoriously difficult to determine the structure of an amorphous molecule, especially a relatively large one, because of the distributions in conformational states that exist, compared to a crystalline form.

The primary focus of this paper is the structural characterization of the PROTAC 2 using solid-state NMR spectroscopy, complemented with molecular dynamic simulations, to fully understand the structure of the PROTAC 2 in the amorphous state.

Noteworthy results:

determining structures of amorphous molecules is challenging, and this represents a next level assignment of such a large molecule.

Significance to field and comparison to literature:

the ability to solve the structure of a molecule this complex in the solid state is something that I would not thought possible a few years ago. it takes a complex set of NMR experiments, including high-resolution proton NMR and sensitivity-enhanced DNP NMR to be able to get the information from the system. the ability to model these systems computationally is also quite helpful, but the combination is needed to extract all of the information needed

literature - the closest work that has been done in this area is references 51 and 54, which includes a previous nature communication on a much less complex system

We thank the reviewer for their very positive appreciation.

Flaws:

Not really a flaw, but I do wish that the authors would comment on the potential conformational distributions that would exist between the material in the powder form vs that of the frozen solution. In particular, there is a potential for structural relaxation to occur in the powder state, resulting in a smaller range of conformations as the material eliminates the higher energy conformations. I would not anticipate seeing this in the frozen state, so it would be interesting to compare the carbon spectrum of the frozen solution to that of the powder and look for any differences.

The reviewer brings up an interesting question, about the structure that might be present in a frozen solution. While this is not directly relevant to the present study, we have added a comparison of the carbon-13 CPMAS spectra for the frozen solution and the as-received powder in new Figure S5. The spectra are quite similar (though not identical), indicating that there is no very significant change, as is probably expected (following the reviewer's reasoning). However, we note that a majority of experimental distributions used to determine the full structure of the powder drug form are determined from the 100 kHz MAS hCH spectrum. To determine a complete structure of the drug in the frozen solution would require an in depth study that is out of scope here.

Methodology:

this is work coming from the leading group in solid-state NMR spectroscopy, and clearly exceeds the standards in the field

Thank you!

Reproducibility:

the work can be easily reproduced, provided that the instrumentation and skills to acquire and

interpret the NMR data were available to do so - that being said, there are very few places in the world where this work could be done

Thanks

Overall comments:

This is an exquisite study of the amorphous structure of a relatively large molecule. The only part of the manuscript that is not of high quality is the introduction. I found it to be poorly written and not motivating the problem particularly well. For example, the sentence "This leads to high interest in the atomic-level structure of amorphous forms, since, in terms of bioavailability, the risk of later structural changes due to environmental factors prevents them from being broadly applicable." I have no idea what the authors are trying to say - is it that the material might crystallize? What structural changes would lead to changes in bioavailability, short of crystallinity? The compound does not appear to crystallize.

We have rearranged the text to make this point clearer:

"Amorphous solids are of high interest as pharmaceutical formulations, especially in the case of complex molecules such as PROTACS which in general do not readily crystallize. However, the risk of later spontaneous structural rearrangements due to the environment, that will change the bioavailability, prevents amorphous forms from being broadly applicable."

Minor comments:

P.2 it is labeled Scheme 1, but should probably be Figure 1

Thank you for spotting this, now corrected.

Reviewer #2 (Remarks to the Author):

Reviewer #3 (Remarks to the Author):

This is a very nicely described and clearly evidenced NMR crystallography study of a large molecule that exhibits both short and long range disorder. The molecular disorder makes this a challenging case for structure elucidation, and the work clearly describes the processes employed. The conclusions are well evidenced through those descriptions.

We thank the reviewer for their very positive appreciation.

The authors could comment on the extent to which the ^{13}C and ^{15}N solid-state NMR line shape dependence on conformational distribution v T2 effects and line broadening due to incomplete averaging of X-1H dipolar coupling affects their analysis and resulting structural parameters they discuss. This is a common issue for NMR in structure determination and this paper could become a point of reference to highlight when the contributions to NMR line shape from factors other than molecular conformation distribution can be ignored and how to test whether that is the case.

This is a very pertinent comment. Here, the ^{13}C distributions typically have widths of 1-3 ppm (250-600 Hz). Today it is well established that in molecular solids, at fast MAS (with properly adjusted decoupling and MAS settings) the residual linewidths (due to incomplete removal of the anisotropic interaction (T_2'), and to magnetic susceptibility broadening) are less than 50 Hz. This is within the errors of the analysis here. We have added a comment to the experimental section to make note of this, with references.

Otherwise the paper is an exemplary study, excellent compilation of NMR parameters, assignment etc in the SI, and methods in the main text, certainly sufficient to be reproduced.

Thanks again for the very positive comments.

Reviewer #4 (Remarks to the Author):

Emsley, Schantz, and coworkers submitted a manuscript reporting the structural determination of an amorphous drug molecule, PROTAC 2. Amorphous solid dosage forms represent a major category in modern pharmaceuticals, yet their molecular-level structural understanding remains highly limited. Conventionally, drug molecular structures are determined in crystalline form by single-crystal X-ray diffraction or in organic solvents using solution NMR, primarily assisting discovery and process chemists during early drug development stages. However, most commercial drug products are formulated in amorphous or partially amorphous forms, and their structures are not meaningfully elucidated in these solid states. This presents a significant knowledge gap in understanding their physical stability, potential instabilities, and functional properties relevant to drug delivery.

Structural determination of amorphous materials remains a pioneering research area, with only a limited number of examples reported to date, particularly in pharmaceutical sciences. This study carries high novelty by utilizing a combination of advanced solid-state NMR techniques (including ultra-fast MAS, DNP enhancement), machine-learning-based chemical shift prediction, and molecular dynamics simulations to address a challenging and important problem. The experimental design and data quality are very strong, even considering the expected broad lines typical for amorphous APIs.

I highly recommend this manuscript for publication. The work is novel, rigorous, and timely. I only provide a few questions and comments below for the authors to consider during revision.

We thank the reviewer for their very positive appreciation.

(1) The authors highlight the role of an intramolecular hydrogen bond (H1-N53), which provides stabilizing energy of approximately 10 kJ/mol. However, compared to potentially stronger intermolecular hydrogen bonding, this stabilization appears modest. Could the authors elaborate on why these stronger intermolecular interactions are not significantly populated? Is this due to kinetic trapping, steric constraints, or entropic penalties associated with organizing multiple chains? As a relevant reference, in a previous study on posaconazole [Lu et al., J. Phys. Chem. B, 2020, 124 (25), 5271-5283], intermolecular hydrogen bonding was shown to play a critical role in stabilizing amorphous drug structures via locally ordered domains. Could the authors comment whether similar local ordering contributes meaningfully to PROTAC 2 stabilization, or whether entropic flexibility dominates?

The reviewer highlights an interesting feature of the structure determined here. For the amorphous formulation of PROTAC2, we find that hydrogen bonding is not the most stabilizing interaction on the bulk scale, and stabilization by H-bonding is localized to the phenol group. We speculate that it is likely that more favorable inter-molecular hydrogen bonds are not formed because stabilization here is dominated by steric interactions. For example, as discussed in the text we see that stabilization of nearly 60 kJ/mol is observed for extended structures compared to compacted structure when looking at the C7-C50 (end-to-end) distance. This likely hints to entropic flexibility being a dominant contribution. However, since this is mere speculation, we do not want to comment further on these aspects in the text.

(2) The analogy made to polymer-like amorphous behavior is very insightful. It would further strengthen the manuscript if the authors could correlate this observation to known polymer glass transitions and conformational entropy contributions, especially in relation to the glass transition temperature of PROTAC 2 and the high flexibility of its linker region.

We have now augmented the discussion in the text to note that that T_g of the sample studied here was $\sim 105^\circ\text{C}$, so that the sample is clearly in the glassy state. We also now note that the T_g is similar

to polymers such as polystyrene (~100 °C) and co-povidone (~110 °C), which also have mixed rigid and flexible parts of their structures.

(Any further discussion beyond this would be highly speculative at this stage.)

(3) There are a few carbons that remain unassigned. Some of these, such as C34 and C36, exhibit characteristic chemical shifts (~35.2 and 37.7 ppm, respectively) from solution NMR data and are likely located in the linker region. Could the authors elaborate on the reasons why these peaks could not be assigned in the solid-state spectra? Since chemical shift assignment is critical for downstream structure determination, it would strengthen the manuscript if the authors could attempt to complete assignments for the remaining resolved peaks, particularly those visible in Figure 2. Furthermore, the authors are encouraged to illustrate the assignments more explicitly by including expanded or annotated spectral plots to complement Figure 2.

The reviewer is correct in noting the critical nature of assignment in the workflow. Note that C34 was indeed assigned (see Table S5). More generally, given the high levels of spectral overlap, we only consider an atomic site to be confidently assigned if it can be cross checked from at least 2 experiments. Failing to meet this criteria, we do not include them in the analysis (e.g. C36, C24). Very importantly, we have previously found (e.g reference Torodii (JACS 147, 20, 17077-17087, (2025)) and reference Holmes (Faraday Discuss. 225, 342-354 (2025))) that the complete assignment is not necessary for complete and accurate atomic-level structure determination.

We have now updated the manuscript with expanded annotated plots, as requested, in Figure S1, and a comment on the partial assignment.

(Further analysis of this very interesting point is out of the scope here. A manuscript describing the relation between the degree of assignment and the accuracy of structure determination for molecular solids is in preparation, and will be published in a more specialised forum.)

Minor comments and suggestions:

(4) The title could be slightly more precise in emphasizing the methodology, for example: "Solid-State NMR Structural Determination of an Amorphous VHL-based SMARCA2 PROTAC."

We prefer the current title.

(5) In the MD simulation section, please clarify whether cis-amide conformations contribute significantly to the final NMR-selected ensemble.

The population of the cis-amide conformations are low compared to the trans- conformations. (See figure S17).

(6) As a suggestion, the authors may consider adding a summary table listing the total number of ¹H, ¹³C, and ¹⁵N sites in PROTAC 2 and how many have been assigned in both solution-state and solid-state NMR. This would help readers appreciate the extent of assignments.

We note that the text includes this sentence:

"In this way, 41 carbon and 31 proton solid-state chemical shifts (*i.e.* all those assigned in solution except C23, C24, C36, C40, C46, C47, H24, H36, H40, H42, and H47) were unambiguously assigned (Table S5)."

(7) In the Introduction section, the authors have included many valuable, but largely non-pharmaceutical, references on the use of ssNMR in disordered materials, along with a few pharmaceutical examples that primarily originate from their own research group. Despite the limited amount of existing literature directly probing the structure and interactions of amorphous pharmaceuticals, several highly relevant studies appear to be missing. For example: Molecular Pharmaceutics, 2016, 13 (11), 3964-3975; Molecular Pharmaceutics, 2019, 16 (6), 2579-2589; Molecular Pharmaceutics, 2020, 17 (7), 2585-2598; Journal of Pharmaceutical and Biomedical

Analysis, 2019, 165, 47-55; Molecular Pharmaceutics, 2020, 17 (6), 2196-2207; The Journal of Physical Chemistry C, 2022, 126 (29), 12025-12037.

Thank you, we have added more references to the introduction as suggested.

One reviewer had two remaining very minor comments:

Comment: The only additional comments that I would make would be to recommend that the paragraph in the introduction beginning "Amorphous solids ..." be cut. The revision does not really address the fundamental pharmaceutical concerns, and the manuscript stands alone just based upon the need to understand the structure of complex amorphous systems.

Reply: We would like to keep this paragraph since (i) it provides needed general context for the non-specialised audience; and (ii) we do actually determine the stabilisation mechanism for the amorphous form here, so we do actually address the pharmaceutical concern!

Comment: As a minor comment, with the addition of the sentence: "We note in passing that the Tg of the sample (~105 °C) is similar to polymers such as polystyrene (~100 °C) and co-povidone (~110 °C), which also have mixed rigid and flexible parts of their structures." it should be noted that only higher molecular weight (30 - 1,000 K) polystyrene has a glass transition temperature around 100 C. Lower molecular weights, e.g. 1K (~40-60C) and 5K (~70-80C), have a much lower glass transition temperature. Just adding "high molecular weight" in front of polystyrene would solve this. Copovidone does not really have different molecular weights in practical use, so no need to add a comment about molecular weight.

Reply: We have added "high molecular weight" as suggested.